# Effects of Zinc Oxide Nanoparticles on Growth, Development, and Flavonoid Synthesis in *Ginkgo biloba*

**DOI:** 10.3390/ijms242115775

**Published:** 2023-10-30

**Authors:** Qingjie Wang, Shiyuan Xu, Lei Zhong, Xiya Zhao, Li Wang

**Affiliations:** College of Horticulture and Landscape Architecture, Yangzhou University, Yangzhou 225009, China; wangqingjie@yzu.edu.cn (Q.W.); mz120211344@stu.yzu.edu.cn (S.X.); mz120211345@stu.yzu.edu.cn (L.Z.); mz120221406@stu.yzu.edu.cn (X.Z.)

**Keywords:** *Ginkgo biloba*, ZnO-NPs, mineral elements, gene expression, *GbF3H* gene

## Abstract

*Ginkgo biloba* is a highly valuable medicinal plant known for its rich secondary metabolites, including flavonoids. Zinc oxide nanoparticles (ZnO-NPs) can be used as nanofertilizers and nano-growth regulators to promote plant growth and development. However, little is known about the effects of ZnO-NPs on flavonoids in *G. biloba*. In this study, *G. biloba* was treated with different concentrations of ZnO-NPs (25, 50, 100 mg/L), and it was found that 25 mg/L of ZnO-NPs enhanced *G. biloba* fresh weight, dry weight, zinc content, and flavonoids, while 50 and 100 mg/L had an inhibitory effect on plant growth. Furthermore, quantitative reverse transcription (qRT)-PCR revealed that the increased total flavonoids and flavonols were mainly due to the promotion of the expression of flavonol structural genes such as *GbF3H*, *GbF3′H,* and *GbFLS*. Additionally, when the *GbF3H* gene was overexpressed in tobacco and *G. biloba* calli, an increase in total flavonoid content was observed. These findings indicate that 25 mg/L of ZnO-NPs play a crucial role in *G. biloba* growth and the accumulation of flavonoids, which can potentially promote the yield and quality of *G. biloba* in production.

## 1. Introduction

Globally, climate change and land degradation significantly impact the sustainability of plant production [1]. In response to this threat, numerous technologies have been developed and upgraded, including new fertilizers, pesticides, and high-yield varieties [2,3]. Nanotechnology has emerged as a highly promising and impactful approach in enhancing agricultural productivity and crop-yield efficiency [4,5]. Nanoparticles possess unique physicochemical properties, including enhanced reactivity, distinct surface structures, and high surface-to-volume ratios [1,6]. These properties enable nanoparticles to be utilized as nanofertilizers, facilitating controlled and targeted release of mineral nutrients to plants, thereby promoting their growth and development [1].

Zinc oxide nanoparticles (ZnO-NPs) are extensively produced and used nanomaterials worldwide with a diameter of 1–100 nm [7,8]. Due to their small size, ZnO-NPs can be absorbed by the stomata and cuticle of root hair cells and plant leaves, entering various plant tissues, and provide a way to enhance plant growth and yield [9,10]. Zinc (Zn) is an essential mineral nutrient for plants, playing a crucial role in various physiological processes such as protein, chlorophyll, and carbohydrate synthesis [11,12]. Compared to traditional ZnO fertilizers, ZnO-NPs possess characteristics such as small size, large specific surface area, easy dissolution, and diffusion, allowing plants to rapidly absorb the Zn released by ZnO-NPs, meeting their nutritional needs and promoting growth and development [13]. For instance, the application of ZnO-NPs has been found to significantly enhance cotton, maize, and wheat growth, which ultimately increases the relative yield of the crop [14,15,16]. Moreover, the application of ZnO-NPs is found to affect the accumulation of flavonoids. In dragonhead, cucumber, and mustard, ZnO-NPs can significantly increase the content of flavonoids [1,17,18].

*Ginkgo biloba* (*G. biloba*) is a significant tree species in China, renowned for its ornamental, edible, and medicinal properties [19]. The leaves of *G. biloba* are abundant in flavonoids and terpenes, known as GbE (*G. biloba* leaf extract), and are commonly employed in treatment of cardiovascular and cerebrovascular diseases [20,21,22]. Several measures have been implemented to improve the yield and quality of *G. biloba* leaves, including different density treatments [23], UVB treatments [24], rejuvenation techniques [25] and hormone treatments [26]. Nevertheless, the impact of ZnO-NPs on the growth and flavonoid content of *G. biloba* remains unknown. Therefore, in this study, we treated *G. biloba* seedlings with different concentrations of ZnO-NPs and observed 25 mg/L ZnO-NPs can promote growth and development and enhance the accumulation of flavonoids. The effect of ZnO-NPs on flavonoid accumulation in *G. biloba* leaves mainly depends on their impact on the expression of flavonoid synthesizing genes. The application of ZnO-NPs technology in *G. biloba* production can be utilized to enhance both yield and quality.

## 2. Results

### 2.1. Characterization of ZnO-NPs

First, the TEM was used to examine the morphology of the ZnO-NPs. The size of the ZnO-NPs is quasi-spherical, with a diameter of approximately 15–30 nm (Figure 1a). In addition, the TEM result also shows that some ZnO-NPs agglomeration is occurring, possibly due to the high specific surface area and surface energy (Figure 1a). Next, the XRD pattern shows distinct peaks that are characteristic of the highly intensified ZnO-NPs. The observed peaks at 31.765°, 34.402°, 36.240°, 47.538°, 56.589°, and 62.841° correspond to the angles (100), (002), (101), (102), (110), and (103) of the standard value (PDF#99-0111). The noticeable broadening of the diffraction spikes signifies a smaller grain size in the ZnO-NPs. The grain size of solid ZnO-NPs samples is estimated to be 24.6 nm using the Scherrer equation (D = 0.89 λ/(β cosθ) (Figure 1b). The XRD diffraction peaks show good intensity, suggesting the excellent crystallinity of the biologically synthesized sample (Figure 1b). In addition, the XPS elemental survey scan spectrum indicates the presence of zinc (Zn), oxygen (O), and carbon (C) elements in the ZnO-NPs sample. A small peak of C was observed due to residual organic matter in the sample. In the Zn 2p spectrum, the fitting peaks at 1021.22 and 1044.21 eV correspond to Zn 2p3/2 and Zn 2p1/2, respectively, confirming the presence of Zn. Furthermore, the high-resolution spectrum of O1s displayed a broad curve that could be deconvoluted into two peaks: the lattice oxygen peak (530.07 eV) and the oxygen vacancy peak (531.74 eV) (Figure 1c).

### 2.2. Effects of ZnO-NPs on G. biloba Growth and Development

To investigate the effects of ZnO-NPs on growth and development, *G. biloba* seedlings were treated with different concentrations of ZnO-NPs. Compared to the control group, the treatment with 25 mg/L of ZnO-NPs promoted the growth of *G. biloba* seedlings, resulting in an increase in the number and growth of lateral roots. However, the application of 50 and 100 mg/L of ZnO-NPs inhibited the growth of *G. biloba* plants, leading to a decrease in both the number and size of lateral roots (Figure 2a). In addition, the plant height and stem diameter of *G. biloba* increased significantly under the 25 mg/L treatment. However, no significant difference was observed between the 50 mg/L and 100 mg/L treatments (Figure 2b,c). The fresh weight of leaves after the 25 mg/L ZnO-NPs treatment was observed to be 1.274 ± 0.07 g, which represented an increase of 12.7% compared to the control treatment. Conversely, the fresh weight of leaves decreased by 29.3% and 39.1% after treatment with 50 mg/L and 100 mg/L of ZnO-NPs, respectively (Figure 2d). Similarly, the dry weight of leaves increased by 31.4% after the 25 mg/L ZnO-NPs treatment, while it decreased by 25.8% and 35.1% after treatment with 50 mg/L and 100 mg/L ZnO-NPs, respectively (Figure 2e).

### 2.3. Effects of ZnO-NPs on Flavonoid Content in G. biloba Leaves

The flavonoid content of *G. biloba* leaves was measured at varying concentrations of ZnO-NPs. The total flavonoid content of *G. biloba* leaves treated with 25 mg/L ZnO-NPs was found to be 23.627 ± 0.383 mg/g, which showed an increase of 16.7% compared to the control group. However, when treated with 50 mg/L and 100 mg/L of ZnO-NPs, the total flavonoid content of *G. biloba* leaves decreased by 7.2% and 11.1%, respectively, with values of 18.794 ± 0.170 mg/g and 17.989 ± 0.134 mg/g compared to the control (Figure 3a). As a result, *G. biloba* leaves treated with 25 mg/L of ZnO-NPs were selected for further analysis of flavonol aglycones. In the treatment with 25 mg/L of ZnO-NPs, the *G. biloba* leaves had an increased content of kaempferol, isorhamnetin, and total flavonol aglycones, measuring 1.37 mg/g, 0.11 mg/g, and 5.91 mg/g, respectively (Figure 3c,d). These values represent increases of 5.9%, 12.4%, and 3.7% compared to the control group. However, the content of quercetin was found to be 0.87 mg/g (Figure 3b), which showed no significant difference compared to the control group.

### 2.4. Effect of ZnO-NPs Treatment on Mineral Elements Content in G. biloba Leaves

The mineral elements in *G. biloba* leaves varied between the treatment with 25 mg/L of ZnO-NPs and the control. The results showed that under the 25 mg/L ZnO-NPs treatment, the Zn and iron (Fe) contents in *G. biloba* leaves were 0.114 mg/g and 0.206 mg/g, respectively. These values increased by 142.6% and 4.6% compared to the control (Figure 4a,b). On the other hand, the copper (Cu), sodium (Na), magnesium (Mg), and potassium (K) contents decreased by 16.7%, 50.7%, 9.6%, and 40.6%, respectively. Additionally, there was no significant difference in manganese (Mn) content under the 25 mg/L ZnO-NPs treatment (Figure 4c).

### 2.5. Effect of ZnO-NPs on Gene Expression in Flavonoid Pathway

Since the treatment of *G. biloba* leaves with 25 mg/L of ZnO-NPs resulted in a significant increase in flavonoid content, qRT-PCR was performed to assess the expression of genes related to flavonoid synthesis, such as flavonol synthase (FLS), chalcone synthase (CHS), flavanone 3′-hydroxylase (F3′H), and flavanone 3-hydroxylase (F3H). The findings indicated that the expression of *GbF3H* (*Gb_12934*), *GbF3′H* (*Gb_02188*, *Gb_11301*, *Gb_30377*), and *GbFLS* (*Gb_14031*) was significantly upregulated following the ZnO-NPs treatment. On the other hand, there were no notable changes observed in the expression of *GbF3′H* (*Gb_16632*, *Gb_32738*), *GbFLS* (*Gb_14028*, *Gb_24242*), and *GbCHS* (*Gb_19002*). Additionally, the expression of *GbF3′H* (*Gb_16037*, *Gb_30412*) was found to be significantly down-regulated (Figure 5).

### 2.6. Overexpression of the GbF3H Gene Resulted in Increased Flavonoid Content

Previous studies have proven that the gene *F3H*, which is responsible for the flavonoid structure, plays a crucial role in the synthesis of kaempferol. When *G. biloba* leaves were treated with 25 mg/L of ZnO-NPs, there was a higher accumulation of kaempferol and an increased expression of *GbF3H* (*Gb_12934*) (Figure 3c and Figure 5a). Based on these observations, the *GbF3H* gene was selected for further study to determine its function. To investigate the role of *GbF3H*, the overexpression vector was transferred into *G. biloba* calli and tobacco (Figure 6a,d). Compared to the control *G. biloba* calli and tobacco, the expression of *GbF3H* was significantly higher in the OE-GbF3H calli and OE-GbF3H tobacco, respectively (Figure 6b,e). Moreover, the total flavonoid content in the OE-GbF3H calli showed a notable increase compared to the control *G. biloba* calli (Figure 6c). Similarly, the OE-GbF3H tobacco also resulted in a significant enhancement in the total flavonoid content (Figure 6f).

## 3. Discussion

ZnO-NPs, with their small size ranging from 1 to 100 nm, are widely utilized in the field of nanotechnology [27]. These nanoparticles have a large specific surface area and exhibit high solubility and diffusion rates, making them easily absorbed by plants [4]. Consequently, the rapid release of Zn^2+^ can regulate physiological, biochemical, and molecular processes that ultimately impact plant growth [27]. Techniques such as TEM, XRD, and XPS are commonly used for the characterization of nanomaterials [28,29]. In this study, TEM, XRD, and XPS spectroscopic analyses showed that the size, properties, and structure of the purchased ZnO-NPs were consistent with previously reported findings (Figure 1) [30,31]. Therefore, these results indicate that the purchased materials possess the structure and properties of ZnO-NPs and can be effectively utilized as novel fertilizers for enhancing plant development and growth.

ZnO-NPs play a crucial role in plant growth, as they can enter the plant system through root hair cells or by being sprayed onto leaves [4,10]. Unlike conventional fertilizers, ZnO-NPs enable the rapid release of Zn^2+^ into plants, leading to enhanced Zn accumulation [13]. In addition, ZnO-NPs carry more nutrients and regulate the content of carbohydrates and other substances in plants, thereby promoting their growth [4]. Numerical studies have shown that ZnO-NPs improve biomass, growth rate, and root growth in clusterbean and pearl millet [32,33]. Moreover, externally applied Zn promotes the accumulation of flavonoid content in plants [18]. There is a close relationship between the nutrient zinc and the state of biomass, which plays a critical physiological and biochemical role in plants. Improvements in plant nutritional status are largely due to changes in root development, which may be a key mechanism contributing to plant nutrition. Lateral root development is known to play an important role in the uptake of fixed elements. Similar to these results, our results showed that ZnO-NPs promoted the accumulation of Zn and altered the changes in root conformation [11]. In this study, the higher Zn accumulation occurred in the treatment with 25 mg/L of *G. biloba* seedlings compared to the control, which indicated that the ZnO-NPs entered the plant and promoted Zn accumulation. Thus, it promoted the growth and development of *G. biloba* seedlings and flavonoid accumulation, highlighting the important role of ZnO-NPs in the growth and development of *G. biloba*. The effects of ZnO-NPs on plants can vary depending on their concentrations. High concentrations can inhibit plant growth, while appropriate concentrations can promote growth and development [27]. For example, when chickpea seedlings were sprayed with a concentration of 1.5 ppm of ZnO-NPs, it significantly promoted root growth. On the other hand, a concentration of 10 ppm had an inhibitory effect on root growth [34]. In this study, we found that concentrations of 50 and 100 mg/L of ZnO-NPs suppressed the growth and development of *G. biloba* seedlings (Figure 2), suggesting their potential toxicity at these concentrations.

Flavonoids are among the most abundant of the phenolic secondary metabolites and are present in all parts of the plant. Flavonoids play an important role in food, medicine, and industry. The flavonoids of *G. biloba* are important medicinal substances and are the raw material of GbE, which can treat cardiovascular and cerebrovascular diseases [20]. ZnO-NPs are involved in the synthesis of secondary metabolites in plants. Previous research has demonstrated that a low concentration of ZnO-NPs can increase the accumulation of total flavonoid content [18]. These results are consistent with the previous report; 25 mg/L of ZnO-NPs significantly boosted the total flavonoid content, particularly flavonol aglycones content (such as quercetin and isorhamnetin), in *G. biloba* leaves. In addition, different concentrations of ZnO-NPs have different changes in the flavonoid content of plants, and in general, higher concentrations of ZnO-NPs will inhibit the accumulation of secondary metabolites. In *Stevia rebaudiana*, concentrations of ZnO-NPs ranging from 0.1 mg/L to 10 mg/L displayed positive effects on the production of total flavonoids. However, when applied at a concentration of 1000 mg/L, ZnO-NPs led to a significant decrease in total flavonoids [35]. However, concentrations of 50 mg/L and 100 mg/L led to a reduction in total flavonoid levels (Figure 3a). The potential toxicity of ZnO-NPs at these concentrations may inhibit the increase in total flavonoid content.

ZnO-NPs are able to influence gene expression and ultimately change the levels of various secondary metabolites. The biosynthesis of flavonoids relies mainly on several key enzymes of the phenylpropanoid pathway. The structural genes involved in the flavonoid biosynthetic pathway, such as *GbFLS*, *GbDFR*, *GbCHS*, *GbF3H*, and *GbF3′H*, have been identified as regulators of flavonoid biosynthesis in *G. biloba* [25,36,37]. In this study, most of the related flavonoid biosynthesis genes were upregulated with 25 mg/L ZnO-NPs treatment, including the *GbF3H* (*Gb_12934*) gene (Figure 5a). F3H plays a key role in the flavonoid pathway and can produce precursors for various classes of flavonoid compounds [38]. To date, *F3H* genes have been identified in *Arabidopsis thaliana* [39], apple [40,41], tobacco, and mulberry [42], where they enhance flavonoid content accumulation. In this study, the transient expression of *GbF3H* resulted in the accumulation of flavonoids in *G. biloba* calli and tobacco (Figure 6). These results indicate that ZnO-NPs enhance the accumulation of flavonoids by regulating the expression of key genes of flavonoid biosynthesis, especially the *GbF3H* gene.

## 4. Materials and Methods

### 4.1. Materials and Characterization

The ZnO-NPs particles (purity 99.5%) were purchased from Nanjing Emperor Nano Materials Co., Ltd. (Nanjing, China). The physicochemical properties of ZnO-NPs particles were characterized by various analytical methods. Firstly, the morphology and size of ZnO-NPs particles were studied by transmission electron microscopy (TEM) (Tecnai 12, Eindhoven, the Netherlands) at 120 kV acceleration voltage. Then, the crystal structure of ZnO-NPs was determined by D8 ADVANCE X-ray diffractometer (XRD) (Bruker AXS, Karlsruhe, Germany). In addition, the elemental composition, molecular structure, and valence state of the ZnO-NPs surface were analyzed by X-ray photoelectron spectroscopy (XPS) using ESCALAB 250Xi system (Thermo Scientific Co., Waltham, MA, USA).

To create the solution, combine 25 mg, 50 mg, and 100 mg of ZnO-NPs with 50 mL of double-distilled water. Use an ultrasonic machine with a frequency of 50 kHz and a power of 300 W to sonicate the mixture for 30 min for optimal dispersion of the nanoscale particles. After sonication is completed, adjust the solution volume to 1 L for future use.

The “Fozhi” variety *G. biloba* seeds used in this experiment were acquired from Pizhou, Jiangsu, China. The soil consisted of peat, coconut bran, vermiculite, and perlite, all of which were purchased from Jiangsu Xingnong Matrix Technology Co., Ltd (Zhenjiang, China). The *G. biloba* seeds were planted in a planting pot and placed in a growth chamber under long-day conditions (25 °C, 16 h light/8 h dark). Four different concentrations of ZnO-NPs (0, 25, 50, and 100 mg/L) were applied as foliar sprays and soil irrigation to 40-day-old *G. biloba* plants every 4 days for 3 treatments. A total of 1 L of solution was used for foliar spraying and soil irrigation for each treatment. Each concentration was applied to 10 *G. biloba* seedlings, and each treatment was repeated three times, resulting in a total of 120 *G. biloba* seedlings being treated. The methodological diagram is shown in Appendix A. The treated materials were immediately placed in liquid nitrogen and stored in a refrigerator at −80 °C.

### 4.2. Determination of Flavonol Aglycones and Total Flavonoids

The dried *G. biloba* leaves (0.2 g) were extracted to determine flavonol aglycones (quercetin, kaempferol, and isorhamnetin) based on previous studies [24]. The resulting extract was subjected to an ultrasonic bath, with each lasting for 30 min. Subsequently, high-speed centrifugation (4 °C, 10,000× *g*) was performed. The flavonol aglycones content in the supernatant was analyzed via high-performance liquid chromatography (HPLC)-mass spectrometry (MS), utilizing an Agilent EclipsePlus™ C18 chromatographic column (150 × 21 mm, 5 µm, Santa Clara, CA, USA). The total flavonoid content of *G. biloba* leaves was determined according to the instructions of a plant flavonoid test kit from Suzhou Comin Biotechnology Co. (Suzhou, China).

### 4.3. Determination of Mineral Element Content

The dried *G. biloba* leaves (0.5 g) were carbonized at a low temperature in an electric furnace. It was then transferred to a high temperature electric furnace and incinerated at 450 °C for 2 h. The resulting ash was dissolved with 2 mL of hydrochloric acid solution (1:1) to ensure complete dissolution, and any impurities were filtered out. Finally, double-distilled water was added, and the volume was adjusted to 25 mL. The mineral elements were measured using a PinAAcle 900F flame atomic absorption spectrophotometer(PerkinElmer Inc., Waltham, MA, USA).

### 4.4. RNA Extraction and Quantitative Reverse Transcription Analysis

Total RNA from the samples was processed using the RNA prep Pure Plant Plus Kit (Tiangen, Beijing, China). To homogenize the samples, liquid nitrogen was used to grind the samples into a powder form, and the operation was carried out on ice to prevent sample degradation. Extraction was then conducted following the instruction manual. After extraction, the RNA was solubilized with 30 μL of RNase-free water, and its integrity and quantity were assessed using a NanoDrop One spectrophotometer (Thermo Fisher Scientific, Waltham, MA, USA). The first-strand cDNA was synthesized following the instructions provided by Vazyme Co., Ltd. (Nanjing, China). The primers were designed using Primer 5.0 software (Premier Biosoft, Palo Alto, CA, USA) and synthesized by Sangon Biotech Co., Ltd. (Shanghai, China) (Appendix A). GADPH was used as the internal reference. Quantitative reverse transcription (qRT)-PCR was conducted using the CFX96TM instrument (Bio-Rad, Hercules, CA, USA) and ChamQ SYBR qPCR Master Mix (Vazyme Co., Ltd., Nanjing, China). The relative expression was determined using the 2^−ΔΔCT^ method in three independent biological replicates.

### 4.5. Functional Verification of GbF3H Gene

The *GbF3H* coding sequences (CDS) from *G. biloba* cDNA were amplified and ligated into the plant expression vector pRI101-GFP (35S::GbF3H). These constructs were then transformed into the *Agrobacterium* strain GV3101. Alongside, the empty vector and 35S::GbF3H were also introduced into induced *G. biloba* calli and tobacco tissues by *Agrobacterium*-mediated transient transformation to obtain control and overexpression (OE-GbF3H) cultures using techniques described in a previous study [26]. The freshly grown and healthy *G. biloba* leaves were selected for the experiment. The leaves were washed with tap water for two hours and then sterilized with 70% (*v*/*v*) ethanol and sodium hypochlorite, respectively. After that, they were washed three times with sterile water. Finally, the leaves were cut into 0.5 cm × 0.5 cm pieces and placed in an induction medium (32 g/L sucrose + 2.215 g/L MS + 4 mg/L NAA + 2 mg/L KT) in order to induce the formation of *G. biloba* calli tissue. Once the *G. biloba* calli tissues were formed (approximately one month later), they were transiently infiltrated with *Agrobacterium* infestation. Approximately four days after the transformation, the expression of the *GbF3H* gene and the total flavonoid content of the *G. biloba* calli tissues were determined. 

### 4.6. Statistical Analysis

The experimental data from both the control group and the treatment groups in this experiment were analyzed using a one-way analysis of variance (one-way ANOVA), followed by Tukey’s post-hoc test for significance analysis. Differences between lowercase letters indicate significance at the *p* < 0.05 level.

## 5. Conclusions

Nanomaterials have a significant impact on the regulation of plant growth, development, and secondary metabolic processes. Specifically, ZnO-NPs play a crucial role in enhancing plant growth and yield. The current research shows that ZnO-NPs have the potential to stimulate *G. biloba* development and increase leaf flavonoid levels. These results obtained from treatments with different concentrations of ZnO-NPs highlight the importance of selecting the appropriate concentration to achieve favorable production outcomes. Additionally, it is suggested that ZnO-NPs may stimulate *G. biloba* growth as a nutrient and influence the expression of flavonoid structural genes, thus enhancing flavonoid accumulation as a signaling molecule. In conclusion, the treatment with 25 mg/L of ZnO-NPs effectively improves the growth and development of *G. biloba* seedlings and induces the accumulation of flavonoids in the leaves by regulating genes that influence flavonoid synthesis.

## Figures and Tables

**Figure 1 ijms-24-15775-f001:**
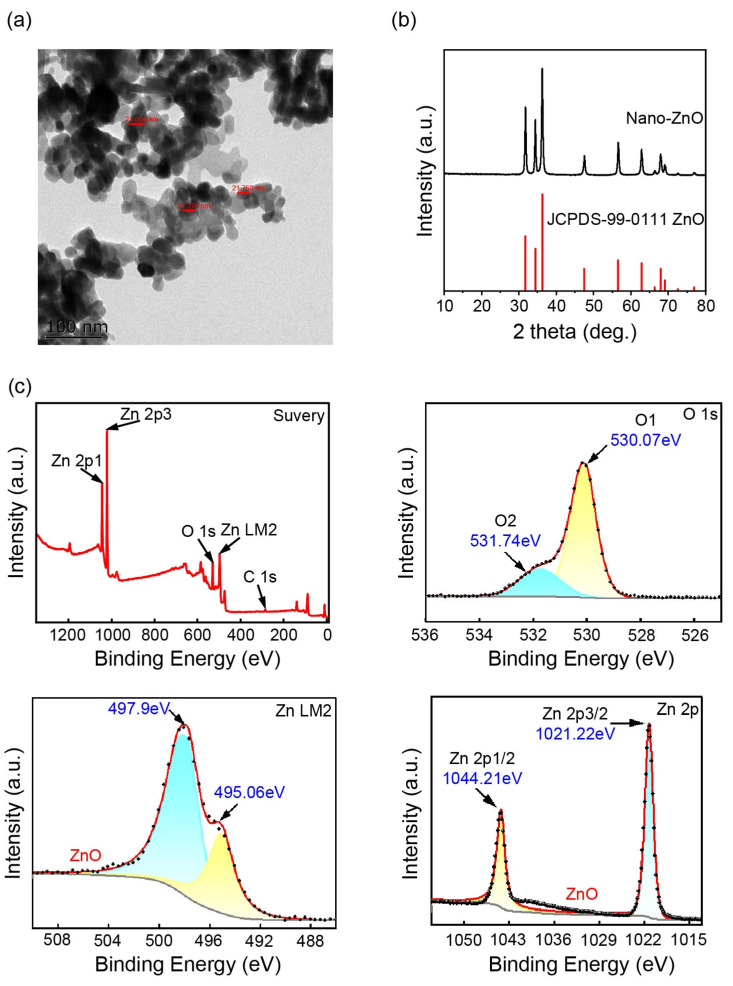
The characterization of the ZnO-NPs by TEM (**a**), XRD (**b**), and XPS (**c**).

**Figure 2 ijms-24-15775-f002:**
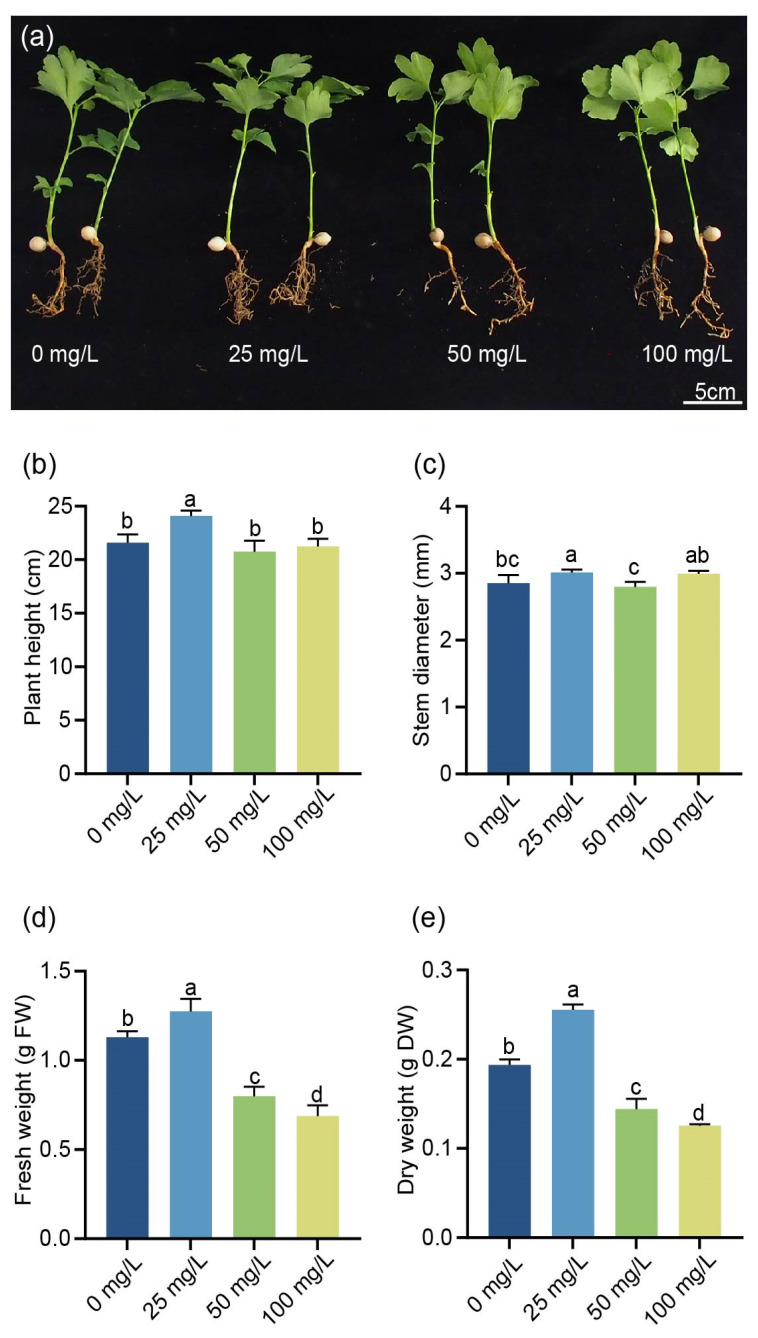
Effects of ZnO-NPs on the *G. biloba* growth and development. Phenotypes of *G. biloba* treated with different ZnO-NP concentrations (**a**); The plant height (**b**), stem diameter (**c**), fresh weight (**d**), and dry weight (**e**) of *G. biloba* seedlings with different concentrations of ZnO-NPs. Means ± SD, *n* = 3. Letters indicate significant differences based on one-way ANOVA (*p* < 0.05).

**Figure 3 ijms-24-15775-f003:**
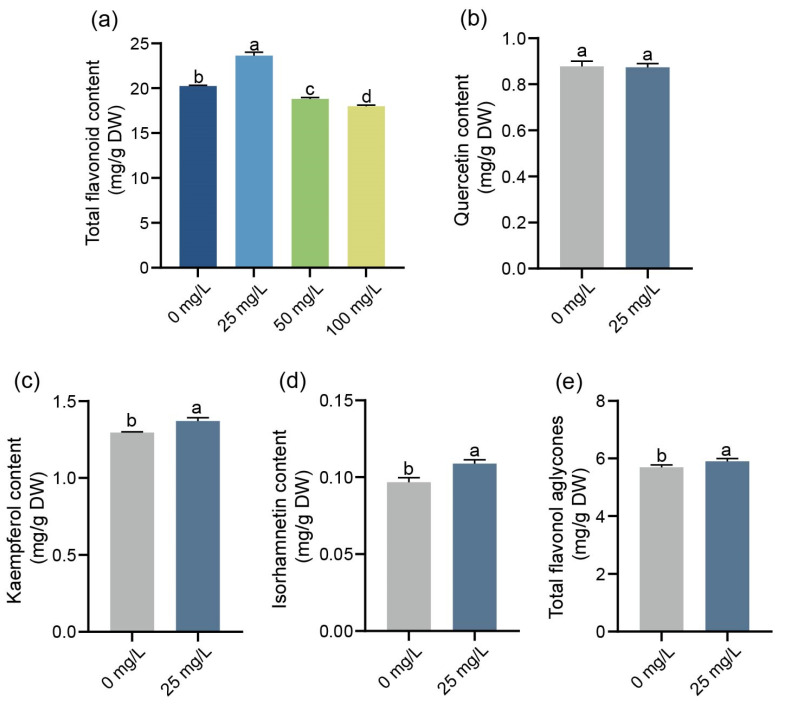
The flavonoid content of the leaves of *G. biloba* under different concentrations of ZnO-NPs. Total flavonoid (**a**), quercetin (**b**), kaempferol (**c**), isorhamnetin (**d**), and total flavonol aglycones (**e**) of *G. biloba* leaves. Means ± SD, *n* = 3. Letters indicate significant differences based on one-way ANOVA (*p* < 0.05).

**Figure 4 ijms-24-15775-f004:**
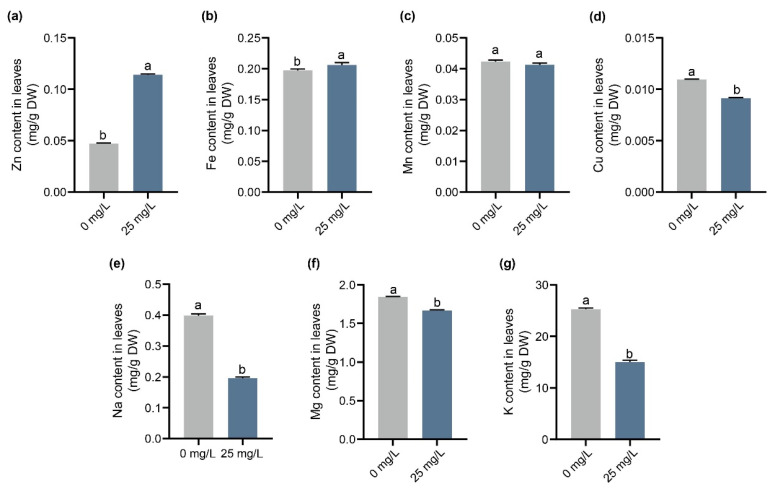
Mineral content of *G. biloba* leaves with 25 mg/L ZnO-NPs treatment. The Zn (**a**), Fe (**b**), Mn (**c**), Cu (**d**), Na (**e**), Mg (**f**), and K (**g**) content in *G. biloba* leaves. Letters indicate significant differences based on one-way ANOVA (*p* < 0.05).

**Figure 5 ijms-24-15775-f005:**
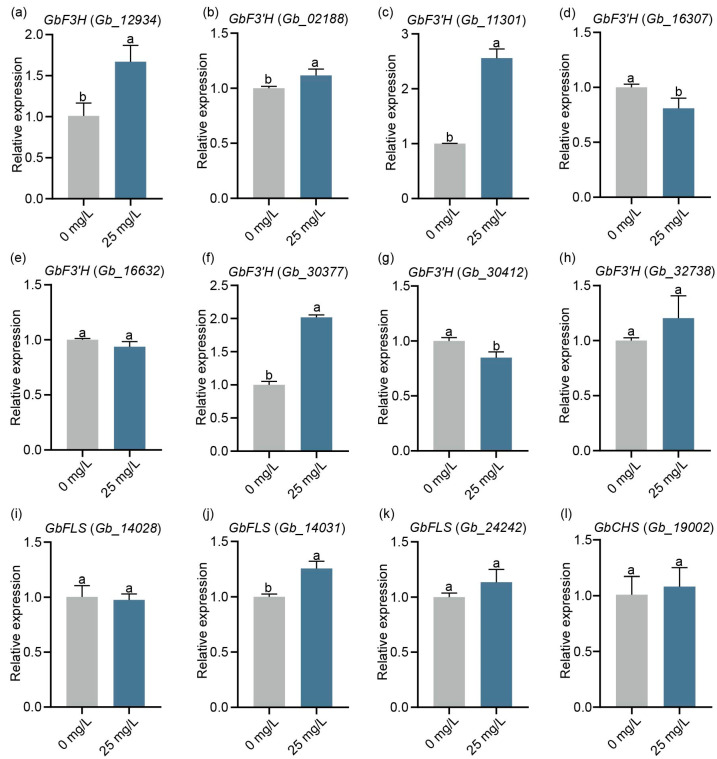
Analysis of flavonoid synthesis genes in *G. biloba* leaves by qRT-PCR. Means ± SD, *n* = 3. Letters indicate significant differences based on one-way ANOVA (*p* < 0.05).

**Figure 6 ijms-24-15775-f006:**
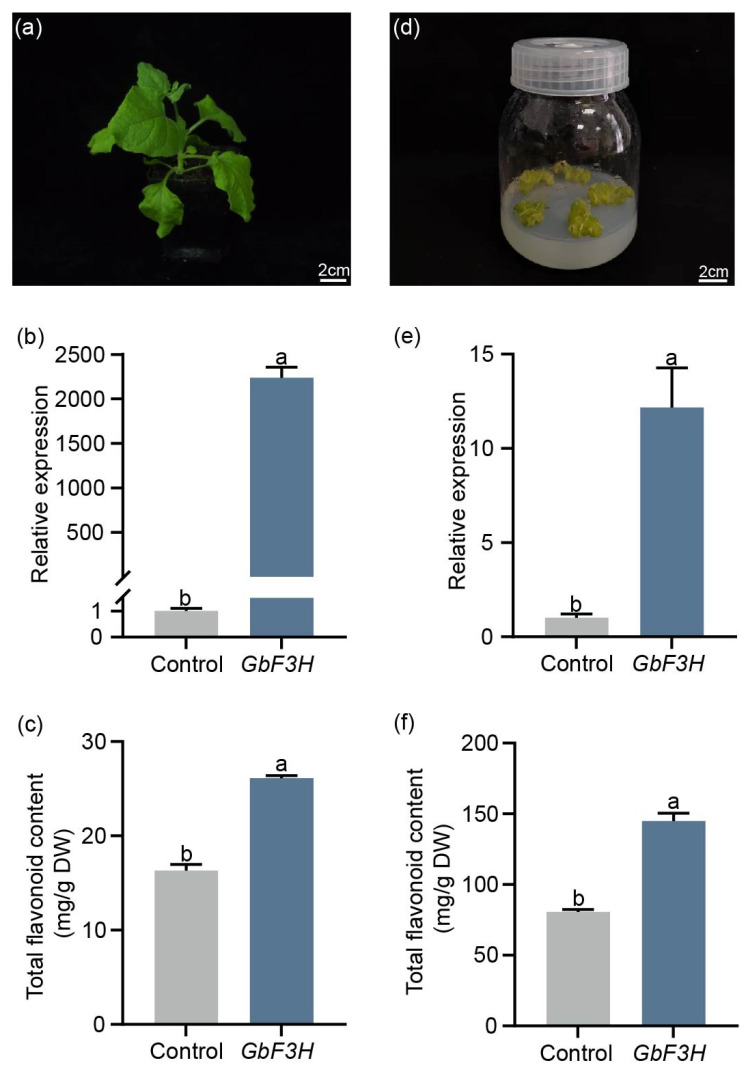
Functional analysis of *GbF3H* gene. Gene expression of *GbF3H* (**b**) and total flavonoid content (**c**) in tobacco (**a**). Gene expression of *GbF3H* (**e**) and total flavonoid content (**f**) in *G. biloba* calli (**d**). Means ± SD, *n* = 3. Letters indicate significant differences based on one-way ANOVA (*p* < 0.05).

## Data Availability

The data are included in the article.

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
