# Peer review of "Effects of Zinc Oxide Nanoparticles on Growth, Development, and Flavonoid Synthesis in Ginkgo biloba"

_ijms, 2023, doi:10.3390/ijms242115775_

Round 1

Reviewer 1 Report

Comments and Suggestions for Authors

I recommend accepting the paper after the authors did specify some experimental details:

1.         How many plants were used for each concentration?

2.         Did authors create samples only foliar spray treatment? Did authors create samples only soil irrigation treatment? Results of these treatments could be interesting, even for results interpretation comparing with compiled treatments.

Author Response

Thank you very much for reviewing our manuscript. We also thank the reviewer for his/her comments and valuable suggestions.

Question 1: How many plants were used for each concentration?

Reply: Sorry for that we did not explain these things clearly. We have changed the sentence as following: "Each concentration is applied to 10 G. biloba seedlings, and each treatment was repeated three times, resulting in a total of 120 G. biloba seedlings being treated" (please see line 259-261).

Question 2: Did authors create samples only foliar spray treatment? Did authors create samples only soil irrigation treatment? Results of these treatments could be interesting, even for results interpretation comparing with compiled treatments.

Reply: That's a good question and thanks for your advice. However, in this manuscript, we do not focus on individual spraying, because spraying in production will inevitably enter the soil, so our experiment is to apply the spraying and the soil together. For the next experiment, we are prepared to do this experiment according to your suggestion.

Reviewer 2 Report

Comments and Suggestions for Authors

The manuscript entitled “Effects of Zinc Oxide Nanoparticles on Growth, Development, and Flavonoid Synthesis in Ginkgo biloba” is well written. I suggest minor revision of manuscript.

Here below are my remarks.

Keywords should not repeat the title. Please, choose the other appropriate words.  For example, seedlings, mineral elements, HPLC-MS analysis, gene expression, GbF3H gene

Page 6 line 130

Moreover, there was no significant difference in manganese (Mg) content under 25 mg/L ZnO-NPs treatment (Figure 4)

There is a mistake. It has a significant difference in manganese (Mg) content under 25 mg/l ZnO-NP (see Figure 4f). The content of Mn is not significantly different from control (see Figure 4c). Please correct it.

Materials and Methods

Page 10 line 264

Please add ( see supplementary file)

Page 11 line 274

Please specify the Agrobacterium strain. Please add more information about G. biloba calli transformation. How is it done? Is a selective nutrient medium with an antibiotic used to select OE-GbF3H cultures

Comments on the Quality of English Language

Minor editing of English language required.

Author Response

Reply to the reviewer

Dear reviewer,

Thank you very much for your positive comments. We have revised our manuscript according to your suggestions.

Reply to the Reviewer 2

Question 1: Keywords should not repeat the title. Please, choose the other appropriate words. For example, seedlings, mineral elements, HPLC-MS analysis, gene expression, GbF3H gene.

Reply: Thank you for your suggestion. We have changed the Keywords as following: " Ginkgo biloba; ZnO-NPs; Mineral elements; Gene expression; GbF3H gene " (please see line 23).

Question 2: There is a mistake. It has a significant difference in manganese (Mg) content under 25 mg/l ZnO-NP (see Figure 4f). The content of Mn is not significantly different from control (see Figure 4c). Please correct it.

Reply: Sorry for the mistake and we have corrected it. We have changed the sentence as following: " Additionally, there was no significant difference in manganese (Mn) content under 25 mg/L ZnO-NPs treatment "(please see line 130-131).

Question 3: Materials and Methods, Page 10 line 264, Please add (see supplementary file).

Reply: Thank you for pointing out this issue. We have changed the sentence as following: " The primers were designed using Primer 5.0 software (Premier Biosoft, Palo Alto, CA) and synthesized by Sangon Biotech Co., Ltd (Shanghai, China) (Table S1). "(please see line 289-291).

Question 4: Page 11 line 274, Please specify the Agrobacterium strain. Please add more information about G. biloba calli transformation. How is it done? Is a selective nutrient medium with an antibiotic used to select OE-GbF3H cultures.

Reply: Thank you for pointing out this issue. We have changed the sentence as following: " These constructs were then transformed into Agrobacterium strain GV3101. Alongside, the empty vector and 35S::GbF3H was also introduced into induced G. biloba calli and tobacco tissues through Agrobacterium-mediated transient transformation to obtain control and overexpression (OE-GbF3H) cultures, using previous study described techniques [26]. The freshly grown and healthy G. biloba leaves were chosen for the experiment. The leaves were washed with tap water for two hours, and then sterilized using 70% (v/v) ethanol and so-dium hypochlorite, respectively. After that, they were washed three times with sterile water. Finally, the leaves were cut into 0.5 cm × 0.5 cm pieces and placed in an induction medium (32 g/L sucrose + 2.215 g/L MS + 4 mg/L NAA + 2 mg/L KT) in order to induce the for-mation of G. biloba calli tissue. Once the G. biloba calli tissues were formed (approximately one month later), they were transiently infiltrated with Agrobacterium infestation. Ap-proximately four days after the transformation, the expression of the GbF3H gene and the total flavonoid content of the G. biloba calli tissues were determined. "(please see line 297-310).

Reviewer 3 Report

Comments and Suggestions for Authors

The Introduction section should clearly delineate the objectives of the paper.

Additional methodological information is required:

  1. Soil Type: It is important to specify the type of soil utilized in the study.
  2. Zinc Concentration: The concentration of zinc employed in the experiments needs to be specified.
  3. Growth Conditions: Please provide details on whether the cultivation took place in a growth chamber under controlled conditions or in greenhouses.
  4. ZnO-NPs Quantity: The quantity of ZnO-NPs used in the experiments must be explicitly stated.
  5. Plant Quantity: Information regarding the number of plants per experimental unit is necessary.
  6. Experiment Repetition: It should be mentioned if the experiment was repeated, and if so, how many times.
  7. Solution Quantity: The amount of ZnO-NPs solution used in a single portion should be specified.
  8. Zn Particles Rate: Clarify the rate of single Zn particles; for instance, 25 mg/L is incorrect as it represents the concentration of the solution, not the quantity in the soil.
  9. Exposure Time: The duration of plant exposure to ZnO-NPs should be indicated.
  10. Methodological Graph: Consider creating a graphical representation of the experimental procedure to illustrate the main steps.
  11. Extraction Procedure Description: A concise description of the extraction procedure should be provided."

Author Response

Reply to the reviewer

Dear reviewer,

Thank you very much for your comments. As suggested, we have revised the sentence in the introduction:" Nevertheless, the impact of ZnO-NPs on the growth and flavonoid content of G. biloba remains unknown." (please see line 238-239)

Reply to the Reviewer 3

Question 1: Soil Type: It is important to specify the type of soil utilized in the study.

Reply: As suggested, we have revised the sentence in manuscript. We added the sentence as following: " The soil consists of peat, coconut bran, vermiculite, and perlite, all of which were pur-chased from Jiangsu Xingnong Matrix Technology Co., LTD. " (please see line 253-254).

Question 2: Zinc Concentration: The concentration of zinc employed in the experiments needs to be specified.

Reply: Thank you for pointing out this issue. We added the sentence as following: "The ZnO-NPs (purity 99.5%) particles were purchased from Nanjing Emperor Nano Materials Co., Ltd." (please see line 239-240).

Question 3: Growth Conditions: Please provide details on whether the cultivation took place in a growth chamber under controlled conditions or in greenhouses.

Reply: As suggested, we added the sentence as following: " The G. biloba seeds were planted in a planting pot and placed in a growth chamber with under long-day conditions (25°C, 16 h light/8 h dark)." (please see line 254-256).

Question 4: ZnO-NPs Quantity: The quantity of ZnO-NPs used in the experiments must be explicitly stated.

Reply: Thank you for pointing out this issue. We added the sentence as following: " Four different concentrations of ZnO-NPs (0, 25, 50, and 100 mg/L) were applied as foliar sprays and soil irrigation to 40-day-old G. biloba plants every 4 days for three treatments. A total of 1 L of solution was used for foliar spraying and soil irrigation for each treatment. Each concentration is applied to 10 G. biloba seedlings, and each treatment was repeated three times, resulting in a total of 120 G. biloba seedlings being treated. " (please see line 256-261).

Question 5: Plant Quantity: Information regarding the number of plants per experimental unit is necessary.

Reply: Thank you for pointing out this issue. We added the sentence as following: " Each concentration is applied to 10 G. biloba seedlings, and each treatment was repeated three times, resulting in a total of 120 G. biloba seedlings being treated." (please see line 259-261).

Question 6: Experiment Repetition: It should be mentioned if the experiment was repeated, and if so, how many times.

Reply: Thank you for your suggestion. We added the sentence as following: " Every concentration treated 10 Ginkgo biloba seedlings, and each treatment was repeated three times, resulting in a total of 120 Ginkgo biloba seedlings being treated." (please see line 259-261).

Question 7: Solution Quantity: The amount of ZnO-NPs solution used in a single portion should be specified.

Reply: Thank you for pointing out this issue. We added the sentence as following: " A total of 1 L of solution was used for foliar spraying and soil irrigation for each treatment. Each concentration is applied to 10 G. biloba seedlings, and each treatment was re-peated three times." (please see line 258-260).

Question 8: Zn Particles Rate: Clarify the rate of single Zn particles; for instance, 25 mg/L is incorrect as it represents the concentration of the solution, not the quantity in the soil.

Reply: Sorry for the misunderstanding. Our study utilized leaf spray and soil irrigation to G. biloba. Additionally, ZnO-NPs were treated thrice in the experiment, thereby making it impossible to ascertain the quantity of ZnO-NPs present in the soil, and consequently, its content cannot be determined.

Question 9: Exposure Time: The duration of plant exposure to ZnO-NPs should be indicated.

Reply: According to your suggestion, we added the sentence as following: " Four different concentrations of ZnO-NPs (0, 25, 50, and 100 mg/L) were applied as foliar sprays and soil irrigation to 40-day-old G. biloba plants every 4 days for three treatments." (please see line 256-258).

Question 10: Methodological Graph: Consider creating a graphical representation of the experimental procedure to illustrate the main steps.

Reply: According to your suggestion, we added the methodological graph in Figure S1.

Question 11: Extraction Procedure Description: A concise description of the extraction procedure should be provided."

Reply: Thank you for pointing out this issue. We added the sentence as following: " To homogenize the samples, liquid nitrogen was used to grind them into a powder form and the operation was carried out on ice to prevent sample degradation. Extraction was then conducted following the instruction manual. After extraction, the RNA was solubil-ized using 30 μL of RNase-free water and its integrity and quantity were assessed using NanoDrop One spectrophotometer (Thermo Fisher Scientific)." (please see line 283-287).

Reviewer 4 Report

Comments and Suggestions for Authors

This is an interesting study comprising, apart from examination the physiological responses of treated with ZnO-NPs Gingko biloba seedlings, also investigation of their effect on gene expressions in context of biosynthesis of flavonoids.

However, there are aspects that require clarification.

Specific comments:

1)     English language requires correction.

2)     Expressions like: we, our us etc. should be removed form scientific text.

3)     Abstract: requires substantial improvements, it should be concise, e.g. it is enough to describe the effect of ZnO-NPs on growth only once instead of three sentences.

Abstract should contain aim/s and the most pronounced results;

4)     Introduction should be shorten and should give the proper ground for the study;

Further:” flavonoid biosynthesis genes may explain the flavonoid accumulation in G. biloba.” – this is obvious, so what exactly, in context of Abstract, was or were aim/aims of the study?

5)     Results:

2.1.                  Why characterisation of ZnO-NPs was performed? Were they synthetised by Authors? Was it one of the aims of the study?

2.2.                  “…resulted in compromised growth..” ? „compromised”?

2.6           “…control EV calli and tobacco…” – what does it mean?

6) Discussion:

- there is no need to repeat results in this section;

7) Materials and Methods:

4.5. – which Agrobacterium strain was used? What was the source of vector used?

7) Conclusions: there is no need to repeat results in this section, just conclude.

Comments on the Quality of English Language

English language requires correction.

Author Response

Reply to the reviewer

Dear reviewer,

Thank you very much for your comments. Now, we have revised the manuscript and made a point-to-point response to address all of the questions and suggestions.

Reply to the Reviewer 3

Question 1:  English language requires correction.

Reply: Thanks for your suggestions. According to your suggestion, we have revised English language in the manuscript.

Question 2: Expressions like: we, our us etc. should be removed form scientific text.

Reply: Thank you for pointing out this issue. We have removed it in the manuscript.

Question 3: Abstract: requires substantial improvements, it should be concise, e.g. it is enough to describe the effect of ZnO-NPs on growth only once instead of three sentences.

Abstract should contain aim/s and the most pronounced results;.

Reply: As suggested, we added the sentence as following: " Ginkgo biloba is a highly valuable medicinal plant known for its rich secondary metabolites, includ-ing flavonoids. Zinc oxide nanoparticles (ZnO NPs) can be used as nano-fertilizers and nano-growth regulators to promote plant growth and development. However, little is known about the effects of ZnO-NPs on flavonoids in G. biloba. In this study, G. biloba was treated with different concentrations of ZnO-NPs (25, 50, 100 mg/L), it was found that 25 mg/L of ZnO-NPs enhanced G. biloba fresh weight, dry weight, zinc content and flavonoids, while 50 and 100 mg/L had an inhibitory effect on plant growth. Furthermore, quantitative reverse transcription (qRT)-PCR revealed that the increased total flavonoids and flavonols were mainly due to the promotion of the expression of flavonol structural genes such as GbF3H, GbF3'H and GbFLS. Ad-ditionally, when the GbF3H gene was overexpressed in tobacco and G. biloba calli, an increase in total flavonoid content was observed. These findings indicate that 25 mg/L of ZnO NPs play a cru-cial role in G. biloba growth and the accumulation of flavonoids, which can potentially promote the yield and quality of G. biloba in production." (please see line 10-22).

Question 4.1:   Introduction should be shorten and should give the proper ground for the study;

4.2 Further:” flavonoid biosynthesis genes may explain the flavonoid accumulation in G. biloba.” – this is obvious, so what exactly, in context of Abstract, was or were aim/aims of the study?

.

Reply 4.1: Thank you for your suggestion. We have shortened and given some reason for the study in the Introduction (please see line 26-61).

Reply 4.2: Yes. Genes related to flavonoid biosynthesis may contribute to the accumulation of flavonoids. However, it remains unknown whether ZnO NPs influence the expression of these genes. Therefore, our investigation aimed to determine whether ZnO NPs affect flavonoid accumulation in G. biloba leaves by altering the expression of specific structural genes involved in flavonoid biosynthesis. These elements are currently unknown and require further exploration in G. biloba.

Question 5:   Results:

2.1.    Why characterisation of ZnO-NPs was performed? Were they synthetised by Authors? Was it one of the aims of the study?

Reply: Thank you for pointing out this issue. The ZnO-NPs utilized in the experiment were purchased from a company; Therefore, the characteristics of the material still need to be assessed. Firstly, it is important to confirm whether the material is indeed ZnO-NPs. Additionally, other properties such as particle size, element type and structure need to be analyzed. It is crucial to consider that nanomaterials of the same kind but with different properties can have a significant impact on the outcome of the experiment. Therefore, it is necessary to examine these properties for our research.

2.2.   “…resulted in compromised growth..” ? „compromised”?

Reply: Sorry for this inappropriate description, and now we have revised it in line 89. " inhibited the growth of G. biloba plants "  

2.6     “…control EV calli and tobacco…” – what does it mean?

Reply: Sorry for this inappropriate description, and now we have revised it in line 157. " Compared to the control G. biloba calli and tobacco "

Question 6: Discussion:

- there is no need to repeat results in this section;

Reply: According to your suggestion, we have removed the repeat results in the Discussion section.

Question 7: Materials and Methods:

4.5. – which Agrobacterium strain was used? What was the source of vector used?

Reply: Thank you for pointing out this issue. The Agrobacterium strain GV3101 was used in this manuscript. The expression vector pRI101-GFP is maintained in our laboratory.

Question 8: Conclusions: there is no need to repeat results in this section, just conclude..

Reply: Thank you for your suggestion. We revised the Conclusions section of the manuscript (please see line 317-328).

Round 2

Reviewer 3 Report

Comments and Suggestions for Authors

The paper was significantly improved and can be published in the present form.

Author Response

Question 1: The paper was significantly improved and can be published in the present form.

Reply:  Thank you very much for your positive comments. 

Reviewer 4 Report

Comments and Suggestions for Authors

The manuscript has been significantly improved, however the species of the Agrobacterium used for experiments has been still not given.

Comments on the Quality of English Language

The grammar should be checked.

Author Response

Question 1: The manuscript has been significantly improved, however the species of the Agrobacterium used for experiments has been still not given.

Reply:  Thank you very much for your positive comments. We have updated our manuscript to include the use of Agrobacterium strain GV3101 in our experiments  (please see line 297).

Question 2: The grammar should be checked.

Reply: According to your suggestion, we carefully checked the grammatical problems throughout the manuscript.